# Harnessing the Potential of Non-Apoptotic Cell Death Processes in the Treatment of Drug-Resistant Melanoma

**DOI:** 10.3390/ijms241210376

**Published:** 2023-06-20

**Authors:** Linyinxue Dong, Ceeane Paul Dagoc Vargas, Xuechen Tian, Xiayu Chu, Chenqi Yin, Aloysius Wong, Yixin Yang

**Affiliations:** 1Wenzhou Municipal Key Laboratory for Applied Biomedical and Biopharmaceutical Informatics, Wenzhou-Kean University, Wenzhou 325060, China; donglinyinxue@wku.edu.cn (L.D.); tianxuechen@wku.edu.cn (X.T.); alwong@kean.edu (A.W.); 2Zhejiang Bioinformatics International Science and Technology Cooperation Center, Wenzhou-Kean University, Wenzhou 325060, China; 3College of Science, Mathematics and Technology, Wenzhou-Kean University, Wenzhou 325060, China; vceeanep@kean.edu (C.P.D.V.); chux@kean.edu (X.C.); 1162990@wku.edu.cn (C.Y.); 4School of Natural Sciences, Dorothy and George Hennings College of Science, Mathematics and Technology, Kean University, Union, NJ 07083, USA

**Keywords:** non-apoptotic cell death, melanoma, anti-cancer therapy, drug resistance

## Abstract

Melanoma is a highly malignant skin cancer that is known for its resistance to treatments. In recent years, there has been significant progress in the study of non-apoptotic cell death, such as pyroptosis, ferroptosis, necroptosis, and cuproptosis. This review provides an overview of the mechanisms and signaling pathways involved in non-apoptotic cell death in melanoma. This article explores the interplay between various forms of cell death, including pyroptosis, necroptosis, ferroptosis, and cuproptosis, as well as apoptosis and autophagy. Importantly, we discuss how these non-apoptotic cell deaths could be targeted as a promising therapeutic strategy for the treatment of drug-resistant melanoma. This review provides a comprehensive overview of non-apoptotic processes and gathers recent experimental evidence that will guide future research and eventually the creation of treatment strategies to combat drug resistance in melanoma.

## 1. Introduction

Melanoma is a type of skin cancer that originates from melanocytes and is responsible for the majority of skin cancer deaths. The incidence rate has been steadily increasing in recent decades. It is also a complex disease that can be caused by a combination of factors such as genetic, environmental, and lifestyle with exposure to UV rays being identified as the main cause [1,2,3,4,5,6]. Melanoma can arise from skin tissues, such as cutaneous melanoma, or non-skin tissues, such as uveal melanoma and mucosal melanoma. Although treatment options including surgery, radiotherapy, chemotherapy, targeted therapy, and immunotherapy have undergone rapid improvements, targeted therapies and immunotherapies have become the prevailing treatments for melanoma patients. Despite promising advancements in these therapies, the emergence of drug resistance poses a significant challenge in achieving long-term treatment efficacy [7].

Programmed cell death (PCD) refers to the physiological elimination of otherwise unneeded or wounded cells within the body, which are further classified into apoptotic and non-apoptotic cell deaths [8,9]. Several non-apoptotic cell deaths have been identified and characterized, including pyroptosis, ferroptosis, necroptosis, cuproptosis, and autophagy-dependent cell deaths. They are characterized by unique individual morphologies and chemical features and are initiated by distinct signaling pathways [10,11,12]. For instance, pyroptosis is triggered by inflammatory signals and mediated by the activation of inflammatory caspases, resulting in cell swelling, plasma membrane rupture, and release of pro-inflammatory cytokines. Unlike necrosis which is a form of accidental cell death, necroptosis is tightly regulated by specific molecular components, mainly the receptor-interacting protein kinases 1 and 3 (RIPK1 and RIPK3) and the mixed lineage kinase domain-like protein (MLKL), which form the necrosome complex and ultimately lead to plasma membrane rupture and necroptotic cell death. However, they both share similar morphological features. Ferroptosis and cuproptosis, on the other hand, are iron-dependent forms of PCD. Ferroptosis is characterized by the accumulation of lipid peroxidation and intracellular reactive oxygen species (ROS) production [13,14], while cuproptosis is a novel copper-induced cell-death mechanism specifically linked to mitochondrial respiration and oxidative phosphorylation [12,15,16].

Although apoptosis is currently the most prominently researched kind of PCD in melanoma development, the dysregulation of apoptosis plays a significant role in the development of intrinsic and acquired resistance observed in melanoma [17,18]. Meanwhile, apoptosis has a proliferative effect on nearby surviving cells known as apoptosis-induced proliferation (AiP), which may contribute to drug resistance [19,20,21]. Recent studies have shown that apoptosis can trigger neighboring surviving cells to proliferate through a paracrine mechanism [19,20]. This finding is of clinical importance as apoptotic treatments can lead to cell proliferation and recruitment of cancer stem cells in different types of cancer, including melanoma.

Given the recent developments in non-apoptotic cell death research in melanoma treatment, this review aims to provide a comprehensive overview of the different non-apoptotic processes and their mechanisms and interactions within the tumor microenvironment, as well as their potential for melanoma therapy. Understanding non-apoptotic cell death mechanisms may yield novel strategies for improving patient outcomes, overcoming drug resistance, and contributing to the identification of new therapeutic targets for melanoma treatment.

## 2. Non-Apoptotic Mechanisms in Melanoma

Understanding the overall mechanisms underlying non-apoptotic cell death pathways in melanoma is crucial to developing novel therapeutic strategies and improving patient outcomes. The mechanisms of pyroptosis, necroptosis, ferroptosis, and cuproptosis are illustrated in Figure 1.

### 2.1. Pyroptosis

Various stimuli can be recognized by intracellular sensor proteins to activate inflammasomes in pyroptotic cell death. For example, melanoma cells can recognize pathogen-associated molecular patterns (PAMPs) from bacteria, such as lipopolysaccharide (LPS), leading to the activation of inflammasomes [16]. Additionally, high-mobility group box 1 (HMGB1), a damage-associated molecular pattern (DAMP) released during melanoma treatment with targeted therapy, can also initiate inflammasomes in melanoma cells [22,23]. Once these signals are sensed, inflammasomes recruit and activate the caspase group, further triggering the cleavage of gasdermin proteins. This results in the formation of pores in the plasma membrane, the release of pro-inflammatory cytokines such as interleukin-1β (IL-1β), and ultimately cell lysis, promoting inflammation and immune response.

GSDME, also known as DFNA5, acts as a downstream executioner of caspase-3, forming N-terminal domains (GSDME-N) that perforate the plasma membrane and induce pyroptosis. Overexpression of caspase-3 and GSDME is often observed in pyroptosis-induced melanoma cells [22,24,25]. Recently, bidirectional regulation between CASP3 and GSDME has been demonstrated, and GSDMD has also been found to play a role in the pyroptosis of melanoma cells [23]. Rogers et al. found that in extrinsic pathways, death receptor ligation activated caspase 8, which further activated caspase 3 and GSDME. GSDME released GSDME-N, which secreted Cytochrome (Cyt *c*) to activate Apaf-1 apoptosome, leading to the further activation of CAPS3, forming a self-amplifying feed-forward loop. Meanwhile, in inflammasome pathways, caspases 1 and 11 are activated, leading to the activation of GSDMD, which further accelerates mitochondria to release Cyt *c* to enhance pyroptosis.

### 2.2. Necroptosis

Necroptosis is initiated by the binding of specific ligands to their respective receptors, such as TNF receptors, TLR receptors, and others [26]. This binding results in the recruitment of several adaptor proteins, forming complex I. Among them, RIPK1 acts as a central switch in regulating cell survival and death pathways, depending on the cellular context and the availability of other signaling factors [27,28]. For example, TGF-β-activated kinase 1 (TAK1) can regulate RIPK1 to promote necroptosis and is associated with disease progression in melanoma [29,30]. When caspase 8 is inhibited, RIPK1 and RIPK3 are recruited to the necrosome, where RIPK3 phosphorylates and activates MLKL [31,32]. MLKL translocates to the plasma membrane, where it forms oligomers that disrupt membrane integrity. This results in the release of cellular contents, such as DAMPs, which trigger inflammation and immune response. Eventually, cell death occurs [33]. In addition, mitochondria dysfunction, lack of energy, and DNA damage can also lead to necroptosis [34].

### 2.3. Ferroptosis

The underlying mechanisms of ferroptosis are complex and involve multiple cellular processes. The accumulation of lipid peroxides (LOOHs) is the direct motivation for ferroptosis, which is generated by lipid peroxidation [35]. The activation of lipid peroxidation can be mainly achieved by the action of the Fenton reaction and suppression of the antioxidant system. The reactive oxygen species (ROS) generated from the Fenton reaction can react with polyunsaturated fatty acids (PUFAs) in cellular membranes, leading to lipid peroxidation. The iron, which participated in this reaction, can be taken up extracellularly via transferrin receptor 1 (TFRC1) and divalent metal transporter 1 (DMT1), or derived from intracellular iron storage, such as ferritin [36]. The antioxidant system plays a crucial role in protecting cells from oxidative damage and hindering ferroptosis. Glutathione (GSH) is a crucial antioxidant molecule that helps to neutralize ROS, while glutathione peroxidase (GPX4) is a selenocysteine-containing enzyme that degrades lipid hydroperoxides (L-OOHs) to their corresponding alcohols (L-OH). The depletion of GSH and inhibition of GPX4 can both lead to ferroptosis. Many chemicals, such as erastin, RSL3, ML162, and ML210, have been proven to selectively induce ferroptosis in melanoma cells and murine models [35,37,38]. Erastin suppresses cysteine intake by targeting the System Xc- transporter through glutathione depletion [37,39]. Unlike erastin, RSL3, ML162, and ML210 directly inhibit GPX4 expression, leading to an increase in ferrous iron and malondialdehyde (MDA) and lipid ROS [40]. It is important to note that individual melanoma cell lines differ in sensitivity to various ferroptosis inducers. For example, RSL3 is much more efficient than erastin in A375 cells, with IC50 values of 3.52 μM and 0.074 μM, respectively [41]. Meanwhile, melanoma cell lines, such as A2058 and C8161, are more sensitive to ferroptosis inducers than A2058, C8161, and G-361 [35,41,42].

Ferroptosis-related key proteins can serve as indicators of melanoma. For instance, system Xc- is responsible for cysteine/glutamate intake as well as GSH biosynthesis [43]. SLC7A11 is the light subunit of system Xc-, responsible for importing cystine, while SLC1A5 is the heavy subunit responsible for exporting glutamate [42]. Meanwhile, GPX4 requires reduced GSH as a cofactor for its enzymatic activity. Therefore, SLC7A11/GSH/GPX4 is an essential axis in melanoma ferroptosis. Leu et al. discovered that the GSH/GSSG ratio could serve as an indicator of the redox status in ferroptosis. Meanwhile, malondialdhyde (MDA) is the most significant end product of lipid peroxidation. By detecting MDA in cell lysates, the lipid peroxidation level can be determined [44]. Gagliardi et al. also identified *CHAC1* as an early ferroptotic marker in melanoma [35].

Several signaling pathways have been deeply understood in melanoma. The Nrf2/HO-1 pathway is an upstream signaling pathway that regulates the Fenton reaction [13,45,46]. By activating the Nrf2/HO-1 pathway, ferroptosis is inhibited due to a reduced Fenton reaction. The same ferroptosis suppressive role of Wnt/β-catenin signaling has also been found in melanoma [47]. Targeting Wnt/β-catenin signaling can exacerbate melanoma ferroptosis by enhancing lipid peroxidation production. In recent years, microRNAs (miRNAs), small non-coding RNAs that regulate gene expression, have been implicated in regulating ferroptosis in melanoma. They are involved in many pathways, including iron metabolism, lipid metabolism, ROS production, and antioxidant defense. The ferroptosis suppressive role of miR-137 and miR-130b-3p has been discovered, while miR-21-3p has been found to promote ferroptosis. Studies have shown that miR-137 negatively regulates ferroptosis by directly targeting glutamine transporter SLC1A5 in vivo and in vitro [42]. MiR-130b-3p targets *DKK1* and activates Nrf2/HO-1 pathways to inhibit ferroptosis in melanoma cells [13]. Mir-9 directly targets glutamic–oxaloacetic transaminase *GOT1* to inhibit lipid peroxidation and iron accumulation, thus suppressing ferroptosis [41]. In contrast, miR-21-3p directly targets *TXNRD1* to facilitate ferroptosis via the potentiation of lipid peroxidation [48].

### 2.4. Cuproptosis

Cuproptosis occurs as one of the dysfunctions resulting from intracellular copper ion accumulation [49]. The initial uptake of copper, in both Cu(I) and Cu(II) forms, is primarily mediated by the protein CTR1 [49,50,51]. The accumulation of copper leads to various dysfunctions within the mitochondrial respiration cycle. For example, copper ions can bind to lipoylated mitochondrial enzymes, such as DLAT, resulting in disulfide bond-dependent aggregation. FDX1 and LIAS are thought to contribute to this protein aggregation, along with the loss of Fe-S clusters. These dysfunctions contribute to oxidative stress, leading to the oxidation of intracellular components such as lipids, proteins, and DNA [12,50,52]. The mutation of P53, which regulates the clearance of ROS as well as DNA damage repair, is thought to play a role in the regulation of cuproptosis [49]. The synergy of oxidative stress, mitochondrial dysfunction, and copper accumulation activates cuproptosis pathways, leading to the release of intracellular contents.

While copper itself is vital to many processes that promote inflammation, the involvement of cuproptosis in inflammation and its potential to trigger an inflammatory response require further research [12,50]. However, genes associated with cuproptosis activation such as *FDX1*, *GCSH*, *LIAS*, *LIPT1*, *PDHA1*, and *PDHB*, may exhibit inflammatory activities. Additionally, pathways related to inflammation such as EMT and KRAS signaling have been found to be enriched in tumors with low cuproptosis index [15]. Current evidence suggests that cuproptosis and its induction may have anti-inflammatory effects.

### 2.5. Common Mechanisms among Pyroptosis, Ferroptosis, Necroptosis, and Cuproptosis

Although pyroptosis, ferroptosis, necroptosis, and cuproptosis present distinct biological processes with unique features and triggers, there are certain commonalities among them.

*Mitochondrial dysfunction plays a key role in triggering non-apoptotic cell death by leading to ROS accumulation.* In ferroptosis, mitochondria dysfunction results in decreased ATP production, iron accumulation, lipid peroxidation, and ROS generation. The production of mitochondrial ROS eventually leads to ferroptotic cell death [36]. In pyroptosis, mitochondrial dysfunction is involved in activating inflammasomes, which triggers the release of pro-inflammatory cytokines and ROS production. ROS production is necessary for activating downstream caspases that cause pyroptosis. Similarly, in necroptosis, mitochondrial dysfunction and oxidative stress lead to ROS accumulation, promoting the activation of RIPK3 and MLKL, which are key mediators of necroptosis [34]. In cuproptosis, mitochondrial dysfunction contributes to the accumulation of copper ions and the subsequent generation of ROS, leading to oxidative stress and cell death. Therefore, ROS homeostasis-related genes may influence multiple forms of cell death. For instance, knocking down *CYGB* in melanoma cell line G361 significantly elevates mitochondrial ROS and results in both ferroptosis and pyroptosis [45]. Additionally, iron-activated ROS accumulation can also lead to pyroptosis [53].

*Cell deaths share common signaling pathways.* The TNF signaling pathway plays a crucial role in mediating cell death processes. TNF-α, a proinflammatory cytokine, upregulates the iron importer transferrin receptor 1 (TFRC) and downregulates the iron exporter ferroportin (FPN) in the ferroptosis process [54,55]. Meanwhile, TNF-α activates NLRP3 in pyroptosis and TNFR1 and RIPK1 in necroptosis, leading to the formation of inflammasomes and necrosome, respectively [27,56]. Consequently, the activation of TNF signaling results in the activation of the MAPK pathway. In brief, the lipid peroxide accumulation and glutathione depletion in ferroptosis, the formation of necrosome, and the activation of NLRP3 inflammasomes all trigger the activation of MAPK signaling pathways. In ferroptosis and necroptosis, JUN and p38 MARK are activated, while in pyroptosis, p38 MAPK and extracellular signal-regulated kinase (ERK 1/2) are activated [22], mediating the production of transcription factors and pro-inflammatory cytokines. Inhibition of TNF and MAPK signaling pathways can attenuate non-apoptotic cell death, making them promising targets for developing novel therapeutics for cancer.

*Some widely recognized biomarkers may not be identical*. HMGB1, for instance, is released and forms pores in the plasma membrane during both pyroptosis and necroptosis [22,57,58]. In the case of ferroptosis induction, HMGB1 can also be released, triggering an immune response [36,59]. Additionally, GPX4, a well-known ferroptosis regulator, has been found to be negatively associated with D-mediated pyroptosis in lethal polymicrobial sepsis [60].

### 2.6. Crosstalks with Apoptosis and Autophagy

In many cases, apoptosis occurs simultaneously with non-apoptotic cell death, and various drugs can induce the coexistence of multiple cell death pathways [31,33,34,61]. PCD is a tightly regulated process, and the interplay between different types of cell death pathways highly relies on caspase. Caspase-8 acts as a molecular switch that determines whether apoptosis or necroptosis occurs [31]. In cells, caspase 8 is activated to trigger apoptosis, while inhibition of caspase 8 can lead to the activation of RIPK3 and MLKL, ultimately resulting in necroptosis. The activation of caspase 8 can further activate downstream caspase 3 directly through the extrinsic death receptor signaling complex or indirectly via a caspase 8-generated truncated Bid (tBid)-mitochondria-apoptosome amplification loop [23]. In the initiation of ferroptosis, caspase 3 can cleave and inactivate the antioxidant enzyme glutathione peroxidase 4 (GPX4), leading to the accumulation of lipid peroxides and ferroptosis [62,63]. Furthermore, caspase 3 cleaves gasdermin, thus, leading to pyroptostic cell death [23]. Therefore, the caspase family plays a crucial role in mediating the crosstalk between apoptosis and non-apoptosis pyroptosis, depending on the cellular context and stimuli.

Studies have shown that autophagy has a dual role in the regulation of ferroptosis in melanoma. Autophagy can promote ferroptosis by facilitating the degradation of ferritin [64,65,66,67]. It can also regulate the expression of genes involved in ferroptosis, such as SLC7A11 and GPX4 [68]. Interestingly, necroptosis and autophagy-dependent cell death are also interconnected, as both involve the activation of the protein kinase RIPK3 and the formation of necrosomes [58,66,69,70]. Moreover, studies have suggested that autophagy can have both pro- and anti-pyroptotic effects. In some cases, suppressing autophagy in melanoma cells can enhance pyroptosis [71]. On the other hand, autophagy can promote pyroptosis by facilitating the formation of inflammasomes or by aiding in the release of inflammatory cytokines [72,73].

## 3. Interactions in the Tumor Microenvironment (TME)

The tumor microenvironment (TME) is an intricate and dynamic milieu comprising cancer cells, stromal cells, an extracellular matrix (ECM), and a diverse array of immune cells, including T cells, natural killer cells, macrophages, dendritic cells, and myeloid-derived suppressor cells, among others. The TME exerts a significant influence on tumor progression, immune responses, drug resistance, and therapeutic outcomes. Non-apoptotic cell death has the capacity to affect the TME in a multifaceted manner, influencing immune cell infiltration, modulating immune response status, and triggering the release of cytokines, among other factors. Conversely, the TME can also induce or inhibit cell death.

However, despite bioinformatics analysis, there is a scarcity of studies investigating the interplay between cuproptosis and TME in melanoma. Further investigation is therefore warranted. In this section, we elucidate the interactions of pyroptosis, ferroptosis, and necroptosis with the TME.

### 3.1. Pyroptosis Strongly Promotes Maturation of DCs and Infiltration of T Cells

Pyroptosis is a well-known form of inflammatory cell death triggered by the activation of inflammasomes. It is characterized by the release of pro-inflammatory cytokines, including IL-1β and IL-18, which trigger an inflammatory response in the TME. The response leads to the recruitment and activation of immune cells, thereby enhancing the anti-tumor effect. Pyroptosis plays a role in promoting the maturation of dendritic cells (DCs), polarization of macrophages, differentiation and activation of T cells, accumulation of natural killer cells (NKs), and reduction in myeloid-derived suppressor cells (MDSCs).

Treatment with the pyroptosis-inducing drug Nano-CD has been shown to significantly increase the secretion of typical secretion substances (TNF-α, IL-12p40, and IFN-γ) by DCs co-cultured with pre-treated B16F10 melanoma cells, with levels up to 18 times higher than non-pyroptosis cells. Moreover, the expression of surface markers on DCs was also significantly elevated by 42–71%, indicating a potent activation of DCs [74]. In the murine melanoma model, induction of pyroptosis using a nano-gel consisting of a BRAF inhibitor and a COX2 inhibitor resulted in a significant increase in the populations of CD^80+^ CD^86+^ DCs and CD^3+^ CD^8+^ T cells within the tumor [75]. Similarly, the combination of a BRAF inhibitor and a MEK inhibitor triggers pyroptosis, leading to increased DC activation and T-cell infiltration in melanoma cells [22]. GSDME-mediated pyroptosis has also been shown to exert a tumor suppressive role by increasing numbers of NKs and T cells while decreasing M2 macrophages and MDSCs [22,75,76].

### 3.2. The Intricate Interplay between Ferroptosis and T Cells

Emerging evidence suggests that ferroptosis has a significant impact on immune cells. It can reshape the immune environment by promoting the maturation of DCs and increasing the infiltration of T cells and (cancer-associated fibroblasts) CAFs, while simultaneously decreasing the presence of M2 tumor-associated macrophages (M2-TAMs), MDSCs, and Tregs [38,48,75,77,78]. Among these interactions, the interplay between ferroptosis and T cells is particularly noteworthy.

CD^8+^ T cells, also known as cytotoxic T cells or killer T cells, play a critical role in cancer immunity. Ferroptosis can trigger T cell infiltration and, in turn, CD^8+^ T cells can induce lipid peroxidation and ferroptosis in tumor cells, thereby enhancing the anti-tumor effect [38,79]. This is attributed to the release of interferon-gamma (IFNγ) by T cells, which downregulates the expression of System Xc-, thereby restraining cysteine uptake [79].

The dysfunction of T cells in TME is known to contribute to poor outcomes in cancer treatment. Despite their robust ability to recognize and eliminate cancer cells, T cells need to be protected from ferroptosis. The accumulation of lipid peroxidation and ferroptosis in CD^8+^ T cells can impair the longevity and antitumor ability of T cells [80]. Inhibition of ferroptosis in CD^8+^ T cells within TME has been shown to restore antitumor activity and improve the efficacy of immunotherapy [81]. Coincidentally, Treg cells require GPX4 to counteract lipid peroxides and prevent ferroptosis in order to maintain T cell activation and anti-tumor immunity [82].

However, a study found that T cells themselves are more sensitive to ferroptosis inducers compared with melanoma tumor cells [83]. Therefore, it is necessary to develop safe ferroptosis inducers that specifically target tumor cells rather than immune cells. Additionally, a deeper understanding of the mechanisms of ferroptosis in both cancer cells and immune cells is required.

### 3.3. Necroptosis Triggers Inflammatory Response

Necroptosis triggers an inflammatory response upon cell rupture, leading to the release of DAMPs and pro-inflammatory cytokines. This release of DAMPs and cytokines plays a role in immune cell activation. For instance, during the induction of necroptosis, high mobility group box 1 (HMGB1) and IFN-γ are released, which contribute to immune cell activation [57,58]. Studies have shown that increasing levels of HMGB1 are associated with the upregulation of expression of MHCII and CD^86^ expression on the surface of macrophages and DCs, indicating their activation [57]. In addition, necroptosis also increases the infiltration of T cells, including CD^4+^ and CD^8+^ T cells, into the tumor microenvironment [84]. Conversely, TNF released by T cells can activate RIPK1-dependent necroptosis in tumor cells [26]. Notably, CAFs can also respond to necroptosis. Treatment with a necroptosis inducer leads to necroptotic cell death in CAFs resulting in potent antitumor immunity [85].

## 4. Addressing Drug-Resistant Melanoma by Targeting Non-Apoptotic Processes

Non-apoptotic cell death shows great potential in overcoming drug resistance in melanoma. The following evidence highlights new therapeutic strategies for resistant melanoma with the induction of non-apoptosis. Current research on non-apoptosis-targeted strategies in the treatment of melanoma is listed in Table 1.

### 4.1. Using Single Agents with High Anti-Tumor Capacity to Target Non-Apoptotic Processes in Cancer Cells

Scientists are actively exploring new treatment strategies that target non-apoptosis cell death pathways to drive therapeutic innovations with rapid response and long-lasting efficiency in melanoma. Many natural components found in food or living organisms have been discovered to have ferroptosis-inducing effects and anti-tumor capabilities. For instance, gambogenic acid, a traditional Chinese medicine extracted from gamboge, has been demonstrated to inhibit the proliferation of melanoma cells both in vivo and in vitro by inducing ferroptosis through p53/SLC7A11/GPX4 pathways [14,68]. Treatment of melanoma cell lines with gambogenic acid resulted in the observation of the ferroptosis-related morphology under transmission electron microscopy (TEM), as well as the accumulation of cytoplasmic lipid reactive oxygen species (ROS) and MDA content. Similarly, nobiletin, a natural product isolated from citrus peel, has shown an anticancer effect by inducing ferroptosis in the SK-MEL-28 melanoma cell line. Nobiletin inhibited cell growth with an IC50 value of 53.63 μM by regulating the GSK3β-mediated Keap1/Nrf2/HO-1 signaling pathway [46].

Other compounds such as gallic acid and hyperforin, well-known antioxidants, have demonstrated the ability to inhibit cell viability and induce ferroptosis in melanoma cells by decreasing GPX4 activity [72,88]. Additionally, engineered ultra-small silica nanoparticles developed by Kim and colleagues effectively induced ferroptosis in melanoma cells and reduced tumor size in melanoma-bearing mice [89].

Several necroptosis inducers are also associated with strong anti-tumor capacity. For example, a novel naphthyridine derivative named 3u has been shown to induce necroptosis in various cancer cells, including melanoma A375 cells [31]. Oregano extract has also been found to trigger necroptosis through mitochondria and DNA damage, inhibiting melanogenesis and cell proliferation [34]. The authors claim that oregano extract is a safe necroptosis inducer, as it revealed no cytotoxicity and mutagenicity in a non-tumor-proliferating cell model. These discoveries highlight the potential of utilizing non-apoptosis cell death pathways, such as ferroptosis and necroptosis, as targets for developing innovative therapeutic approaches in melanoma treatment.

### 4.2. Activating Potent Response and Causing Immunogenic Cell Death

As discussed in Section 3, non-apoptotic cell death has a strong capacity to interact with immune cells. Non-apoptotic cell death pathways have been shown to induce immunogenic cell death and trigger a robust immune response, which can potentially enhance the efficacy of immunotherapies.

In one study, a self-assembling nano-toxin called T22 peptidic ligand (T22-PE24) was developed, specifically targeting CXCR4. This nano-toxin inducted location-dependent and organelle-specific pyroptotic cell death in melanoma cells [90]. In vivo experiments demonstrated that T22-PE24 has anti-cancer effects and minimizes systemic side effects. Significantly, combined treatment of T22-PE24 and PD-1 antibody further enhanced anti-tumor efficacy compared with T22-PE24 or the PD-1 antibody alone. The combined treatment group showed improved tumor inhibition, achieving 60% tumor eradication in B16-luci-bearing C57 mice. The treatment also resulted in a significant increase in the numbers of CD^4+^ and CD^8+^ in the blood and tumor tissue samples. Another study utilized glutathione-responsive nanogels (cited as CDNPs) to induce pyroptosis in murine models [75]. These nanogels contained dabrafenib (BRAF inhibitor) and celecoxib (COX2 inhibitor) and effectively triggered pyroptosis. Combining the nanogels with PD-1 antibody led to significant induction of tumor growth and prolonged survival in melanoma-bearing mice.

Pyroptosis has also inspired gene editing technology. A cooperative Nano-CRISPR scaffold (Nano-CD) was designed with a selected sgRNA to cause immunogenic pyroptosis [74]. Nano-CD induced endogenous GSDME expression and released cisplatin, resulting in pyroptosis in melanoma cells. This pyroptosis acted in its immunosuppressive role by releasing tumor-associated antigens and amplifying the adaptive anti-tumor immune cascade. Notably, this Nano-CRISPR scaffold platform utilized the tumor’s self-produced bioactive protein to trigger pyroptosis in tumor cells without the need for exogenous protein delivery or intricate modifications. When combined with immune checkpoint blockade therapy, Nano-CD inhibited the recurrence and metastasis of malignant melanoma and exhibited systemic anti-tumor immune responses and long-lasting immune memory effect.

Combining a ferroptosis inducer with immunotherapy can also enhance the anti-tumor capacity. A study demonstrated that the joint treatment of erastin with oncolytic virus (OV)-mediated cancer therapy resulted in a synergistic effect [38]. Erastin induced cytotoxicity on melanoma cells via ferroptosis but failed to generate productive and active antitumor immunity. However, co-treatment with OV and erastin improved the efficacy of OV and increased the infiltration of immune cells.

Furthermore, targeting ferroptosis-related genes or signaling pathways can further enhance the performance of immunotherapy. The gene *TXNRD1*, associated with lipid peroxidation generation in ferroptosis, was found to be a key gene [48]. MiRNA-21-3p was found to directly target *TXNRD1* to induce ferroptosis, thereby improving the outcomes of anti-PD-1 immunotherapy in melanoma [48]. Wnt/β-catenin signaling was also proven to regulate melanoma ferroptosis by increasing lipid peroxidation production [47]. ICG001 is a Wnt inhibitor that can enhance the effectiveness of anti-PD-1 immunotherapy by facilitating ferroptosis [47]. The introduction of ferroptosis improved the response to immunotherapy as well [77].

Nano-delivery systems can improve the efficiency of ferroptosis-targeted drugs and further enhance treatment outcomes. For example, researchers have engineered nanoparticles specifically designed for tumor-specific delivery of ferroptosis-inducer RSL3 [78]. This approach not only enriched the immune activity but also sensitized tumor cells to ferroptosis. When combined with PD-L1 antibody, tumor growth was effectively inhibited in murine models. Another study found that a nano-drug carrying an exosome inhibitor (GW4869) and a ferroptosis inducer increased the response of the PD-L1 checkpoint inhibitor in murine melanoma models [77]. Similarly, Ma et al. developed a cancer nanovaccine called Fe@OVA-IR820, which induced ferroptosis and possessed photothermal properties [36]. The authors claimed that it was a safe ferroptosis inducer with low therapeutic toxicity and no side effects on the immune system. Combining Fe@OVA-IR820 with the CTLA-4 checkpoint blockade showed promising results in enhancing anti-cancer immunity and inhibiting tumor growth. Hu et al. conducted research on a *FAP* gene-based exosome-like nanovesicle cancer vaccines called eNVs-FAP [99]. This vaccine exerted its anti-tumor ability by promoting a robust immune response and triggering ferroptosis.

Necroptosis, another form of non-apoptotic cell death, can also trigger an aggressive immune response and enhance the efficacy of immunotherapy. The small molecule curaxin CBL0137 directly targets *ZBP1*, whose expression is associated with RIPK3 and MLKL. By inducing ZBP1-dependent necroptosis in cancer-associated fibroblasts, CBL0137 drives the recruitment of CD^8+^ T cells into tumors and strongly potentiates immunotherapy responses in vivo [85]. Treating melanoma cells with miRNA coding for the necroptosis mediator MLKL can also induce necroptosis and elicit a potent anti-tumor response with increased infiltration of immune cells [84]. The combination of a PD-1 inhibitor with MLKL mRNA was proved effective in suppressing tumor growth in vivo. Furthermore, AZD1775, a WEE1 kinase inhibition, was found to induce necroptosis and sensitize tumor cells to anti-PD-1 therapy by restoring the killing capacity of cytotoxic T-lymphocyte [91].

### 4.3. Synergizing with Other Therapies to Enhance Treatment Efficacy

In addition to immunotherapy, non-apoptosis-targeted treatment can be combined with other cancer therapies, such as targeted therapy, chemotherapy, and photodynamic therapy, to maximize treatment efficacy. Synergistic effects have been observed when combining different agents that induce non-apoptotic cell death.

For instance, the combination of a heat shock protein 90 (Hsp90) inhibitor and a pan-PI3K inhibitor has been shown to synergize in inducing pyroptosis [92]. In vivo and vitro studies found that the combination of AUY-922 (Hsp90 inhibitor) and PI-103 (PI3K inhibitor) induced more pyroptosis than either agent alone. The combination of ferroptosis-targeting agents with targeted therapies has also been explored in melanoma. For example, one study identified the ferroptosis-inducing role of the AXL inhibitor BGB324 in melanoma cells. When it was combined with the BRAF inhibitor vemurafenib, more enhanced tumor inhibition was discovered through the induction of ferroptosis [93].

Photodynamic therapy (PDT), a non-invasive and highly selective cancer treatment modality, has been studied in melanoma treatment. It involves the use of a photosensitizer activated by light to generate ROS, leading to localized cytotoxicity in tumor cells. However, the efficacy of PDT in advanced melanoma still faces challenges that need to be addressed. It is worth exploring whether combining ferroptosis-targeted strategies with PDT can overcome the limitations.

Researchers have synthesized a potent mitochondria-localized photosensitizer called cyclometalated Ir(III) complexes Ir-pbt-Bpa, which exhibits a strong antitumor impact on melanoma cells by inducing ferroptosis and restraining tumor growth in murine models [59]. Another study constructed a nanoparticle-based material named protoporphyrin IX-based polysilsesquioxane platform (PpIX-PSilQ NPs), which synergizes with PDT to mainly induce ferroptotic cell death by upregulating lipid peroxides and inactivation of GpX enzymes [94]. Hence, the use of combined ferroptosis-targeted strategies may provide alternative approaches in designing PDT to improve treatment outcomes.

Elesclomol, an effective cuproptosis inducer, showed incredible synergy with paclitaxel, a chemotherapy drug, in phase II clinical trials for stage IV metastatic melanoma patients [86]. Patients who received the combination treatment of elesclomol and pacilitaxel experienced a significant increase in medium progression-free survival (PFS) and overall survival (OS) with an acceptable level of toxicity. However, limitations of the paclitaxel and elesclomol combined therapy were discovered in phase III, leading to the termination of the trial due to an imbalance in total deaths among patients with low baseline levels of lactate dehydrogenase (LDH) [100]. Although the study did not achieve a significant improvement in PFS (hazard ratio, 0.89; *p* = 0.23), a subgroup analysis revealed a positive outcome for patients with normal LDH, suggesting that LDH may serve as a predictive factor for the efficacy of this combination therapy.

### 4.4. Sensitizing Resistant Cells to Existing Therapy

With the induction of non-apoptosis, resistant melanoma cell lines can be sensitized again to treatment. Studies have shown that melanoma cells deficient in GSDME-mediated pyroptosis fail to mount a durable anti-tumor immune response to BRAFi + MEKi treatment. These resistant cells lacking pyroptosis markers exhibit decreased intra-tumoral T cells and DCs infiltration [22]. However, re-inducing pyroptosis in these cells via chemotherapy drugs such as etoposide or doxorubicin in tumor-bearing mice inhibits tumor growth and significantly improves overall survival.

Furthermore, the IRF9-STAT2 signaling pathway has been found to exacerbate adaptive resistance to BRAF inhibitors by controlling pyroptosis [101]. *IRF9* and *STAT2*, which are part of the Jak-STAT pathway, play a major role in BRAFi resistance in melanoma. Knocking down *IRF9* or *STAT2* in melanoma cell lines A375 and SK-MEL-28 induces GSDME-dependent pyroptosis and restores sensitivity to BRAFi. Targeting IRF9/STAT2 may offer a promising strategy to prevent melanoma resistance to BRAFis by regulating pyroptosis, leading to the durable regression of melanoma in an immune-mediated manner.

In addition, a compound called “Raptinal” has been investigated for its effects on melanoma cells. Raptinal not only inhibits cell proliferation and induces cell cycle arrest but also triggers pyroptosis [24]. It efficiently and rapidly initiates pyroptosis in both naïve melanoma cells and BRAFi-plus-MEKi-resistant melanoma cells, exhibiting great potential in overcoming acquired resistance toward targeted therapies.

Another study discovered that BRAFi-resistant melanoma cells controlled the dysregulation of lipid metabolism pathways [62]. By targeting *ACAT2* or inhibiting *SOAT* using avasimibe in lipid metabolism pathways, the sensitivity to PLX4032 (BRAFi) was restored, and the ferroptosis was increased. This suggests that manipulating lipid metabolism pathways can overcome BRAFi resistance and enhance ferroptosis. Similarly, Chang et al. discovered the ferroptosis-inducing capacity of Phyto-sesquiterpene lactone (DET) and its derivative, DETD-35 [87]. These compounds directly inhibit the GPX4 enzyme and were able to sensitize BRAFi-resistant cells to vemurafenib. Meanwhile, a synergistic effect between lorlatinib and ferroptosis-inducer was found in melanoma [95]. Lorlatinib sensitizes melanoma to ferroptosis through the PI3K/AKT/mTOR signaling pathway, while ferroptosis-inducer significantly improved the anti-tumor effect of lorlatinib in vivo and in vitro. These findings demonstrate the importance of targeting the lipid metabolism pathway, utilizing GPX4 inhibitors, and exploring synergistic effects with other compounds to overcome resistance and enhance ferroptosis in melanoma treatment.

### 4.5. Delaying the Development of Drug Resistance

Targeting non-apoptotic cell death can be an effective strategy to delay the development of drug resistance. Current research has shown that targeting *PDPK1* and *MEK* together can deactivate both PI3K-AKT and MEK-ERK1/2 pathways, resulting in pyroptotic cell death. This is supported by the observation of morphological features indicative of pyroptosis, increased annexin-V and propidium iodide positivity, cleaved caspase-3 and GSDME expression, and enhanced release of HMGB1. In a xenograft model and immune-competent allograft models, tumor growth could be significantly delayed through the joint inhibition of *PKPK1* and *MEK* [25].

Similarly, activation of lipid peroxidation through X-rays and hyperoxia, facilitated by nano-enabled photosynthesis, induced a high level of ferroptosis. This potent induction of ferroptosis demonstrated a synergistic killing effect when combined with dacarbazine or vemurafenib in melanoma cells [18]. In vivo studies using this combined treatment showed delayed tumor growth and hindered development of drug resistance to chemotherapy and targeted therapy.

### 4.6. Targeting Cancer Stem Cells

Cancer stem cells (CSCs) are a small population of cells within a tumor that possess stem-cell-like properties, including self-renewal and the ability to differentiate. CSCs are thought to play a critical role in cancer progression, metastasis, and drug resistance. Targeting necroptosis in cancer stem cells has emerged as a promising approach for treating apoptosis-resistant melanoma [96]. Plasma-activated infusion (PAI), a novel, non-thermal atmospheric pressure plasma-based anti-neoplastic agent, has been shown to induce necroptosis effectively. Salinomycin is an anticancer stem cell agent, which synergizes with PAI by inducing collapse in the mitochondrial network collapse. The combination of PAI and Salinomycin leads to reduced tumor growth.

In addition, scientists also identified a novel Pleuromutilin derivative called compound 38 as an anticancer drug candidate [97]. This compound has demonstrated the ability to significantly increase ROS levels in melanoma cancer stem cells and consequentially induce cell death via necroptosis. In murine melanoma models, compound 38 has exhibited low cytotoxicity while effectively suppressing tumor growth. These findings highlight the importance of targeting necroptosis in cancer stem cells as a means to overcome apoptosis resistance in melanoma. The use of innovative approaches such as plasma-activated infusion and the development of novel compounds such as compound 38 offer promising strategies for specifically eliminating cancer stem cells and improving treatment outcomes in melanoma.

### 4.7. Building Prognosis Models to Predict Drug Efficiency and Clinical Outcomes

Recent studies have shown the significant association between a high level of non-apoptosis cell death and favorable clinical outcomes and treatment responses in melanoma patients. In the past few years, scientists have donated their efforts to build prognosis models based on pyroptosis-related genes (PRGs), ferroptosis-related genes (FRGs), necroptosis-related genes (NRGs), and cuproptosis-related genes (CRGs). These models serve as powerful tools for predicting overall survival (OS) in melanoma patients. For instance, a prognosis model based on 12 PRGs has shown powerful diagnostic and prognostic capabilities for skin melanoma [102]. The model exhibited high area under the curve (AUC) values for predicting 1-, 3-, and 5-year OS, with values of 0.850, 0.822, and 0.898, respectively.

Moreover, these prognosis models could also help to identify patients who are more likely to develop drug resistance or have poor responses to treatment. A study revealed that a group of patients with high pyroptosis scores had a significant survival advantage and a higher immunotherapy response rate [103]. Through gene set enrichment analysis (GSEA) and cell line verifications, the study also discovered that melanoma patients with elevated levels of pyroptosis-related genes may benefit from cisplatin and imatinib [102].

Prognosis models based on non-apoptotic cell death pathways offer clinicians the ability to personalize therapy and optimize treatment regimens for melanoma patients. Additionally, these models can be utilized in preclinical and clinical trials to evaluate the efficacy of new drugs and treatment strategies, facilitating the identification of more effective treatments against resistant cancer cells.

## 5. Future Perspectives

### 5.1. Considering Multiple Forms of Cell Death as a Whole

The resistance of melanoma to treatment poses a significant challenge in its management. Cancer cells employ various mechanisms to escape cell death, including resistance to apoptosis and ferroptosis. For instance, melanoma cells have already been shown to develop resistance to ferroptosis [35,37,44]. Traditional therapies that target one single cell death pathway have accelerated the development of resistance; however, the development of therapeutic strategies that target multiple cell death pathways could circumvent the problem of drug resistance. Alternatively, the creation of new-generation drugs that target a key protein common to multiple cell death pathways could also enhance efficacy. Therefore, highlighting the importance of targeting multiple cell death pathways could increase the chances of successful treatment and reduce the risk of resistance. Combination therapies that involve non-apoptotic cell death inducers with anti-cancer drugs have emerged as potential solutions to overcome drug resistance. For example, cocktail therapy that involves the targeting of multiple pathways with a combination of drugs or treatments is particularly promising in cancers such as melanoma that are resistant to single-agent therapies.

Considering the complex and dynamic networks created by the crosstalk between different forms of cell death, it is crucial to understand and consider all forms of cell death as a whole in therapy development. The concept of PANoptosis, which involves the simultaneous activation of pyroptosis, apoptosis, and necroptosis, has shown significant potential [104]. Strategies targeting PANoptosis have been explored and showed promising outcomes [61,105].

### 5.2. The Mechanisms of Non-Apoptotic Cell Death in Melanoma Need to Be Further Clarified

While the fundamental principles of non-apoptosis are similar across different types of cancer, there may be specific differences in the key genes, proteins, and signaling pathways involved in melanoma. For example, GSDM family members such as GSDMA, GSDMB, GSDMC, GSDMD, and GSDME have been extensively studied for their roles in pyroptosis in various cancers, and the caspase-3/GSDME-dependent pyroptosis signaling pathway plays a dominant role in melanoma. Further research is needed to elucidate the precise mechanisms and unique characteristics of non-apoptotic cell death in the molecular landscape of melanoma or its different subtypes.

### 5.3. The Mechanisms of More Forms of Cell Death Are Waiting to Be Explored

In addition to the cell deaths mentioned above, scientists have identified other forms of cell death, such as disulfidptosis, which is triggered by an imbalance in cellular redox homeostasis [106] However, the underlying mechanisms and therapeutic implications of disulfidpotosis still call for further investigation. Future research aimed at understanding the molecular mechanisms underlying new or existing forms of cell death will undoubtedly provide new insights into the development of potential bookmarks and novel therapeutic strategies.

Overall, considering multiple forms of cell death, their interplay, and uncovering the mechanisms of both known and newly discovered forms of cell death will contribute to a better understanding of cellular signaling complexity and pave the way for the development of effective therapeutic approaches in melanoma and other cancers.

## 6. Conclusions

Non-apoptotic cell death has emerged as a promising therapeutic approach in melanoma treatment. Four prominent forms of non-apoptosis cell death, namely, ferroptosis, pyroptosis, necroptosis, and cuproptosis have been extensively characterized by distinct mechanisms and pathways. The complex and interconnected nature of pyroptosis, ferroptosis, necroptosis, and cuproptosis has been illustrated, as well as their crosstalk with autophagy and apoptosis. They have exhibited great promise in prognostic prediction as well as therapeutic strategies. These forms of cell death are able to overcome drug resistance and have shown promising results in preclinical studies. Future research focused on unraveling the molecular mechanisms underlying these processes will undoubtedly provide valuable insights into the development of innovative therapeutic strategies for the treatment of melanoma. A deeper understanding of how non-apoptosis processes operate in melanoma will contribute to the identification of novel targets and the optimization of treatment regimens, ultimately improving patient outcomes in melanoma therapy.

## Figures and Tables

**Figure 1 ijms-24-10376-f001:**
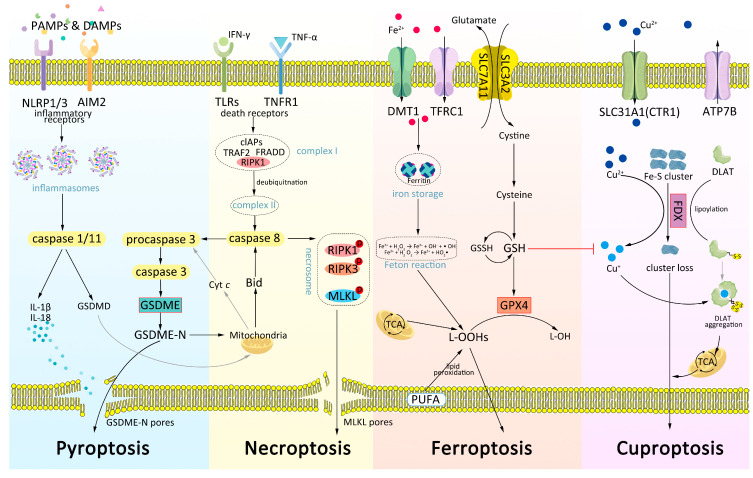
The mechanisms of pyroptosis, necroptosis, ferroptosis, and cuproptosis in melanoma cells.

**Table 1 ijms-24-10376-t001:** The non-apoptotic therapeutic strategies in melanoma.

Drug Name	Drug Type	Cell Death Forms	Cell Lines	In Vivo	Reference
**Clinical Trials**					
Elesclomol-Cucl2	Cuproptosis inducer	Cuproptosis	M619	No	[86]
**Preclinical studies**					
ML162	GPX4 inhibition	Ferroptosis	A375, A2058, B16F10	Yes	[47,87]
ML210	GPX4 inhibition	Ferroptosis	A375, A2058, B16	Yes	[40,83]
IFN-γ	Cytokine	Ferroptosis	A375, A2058, WM793B	Yes	[48]
Gambogenic acid	Natural compound	Ferroptosis + autophagy	A375, A2058, B16, B16F10	No	[14,68]
Nobiletin	Natural compound	Ferroptosis	SK-MEL-28	No	[46]
Gallic acid	Natural compound	Ferroptosis + apoptosis	A375	No	[72,88]
Hyerforin	Natural compound	Ferroptosis + autophagy + apoptosis	A375, FO1, SK-MEL-28, M3Wo	No	[72]
α-MSH-PEG-C’	Silica nanoparticles	Ferroptosis	M21	Yes	[89]
3u	Naphthyridine derivative	Necroptosis (in low concentration), apoptosis (in high concentration)	A375	No	[31]
Oregano	*Origanum vulgare* L. hydroalcoholic extract	Necroptosis + apoptosis	A375, B16F10	No	[34]
T22-PE24	*CXCR4* antagonist	Pyroptosis	A375, A2058, HMCB, ME4405	Yes	[90]
CDNPs	Nanogels loaded with dabrafenib (BRAFi) and celecoxib (COX2i)	Pyroptosis	B16F10	Yes	[75]
Nano-CD	Nano-CRISPR scaffold	Pyroptosis	A375, B16F10	Yes	[74]
Erastin	System Xc inhibition	Ferroptosis	A375, G-361	Yes	[13,41,42]
			A375, A2058, WM793B	Yes	[48]
Erastin	System Xc inhibition	Ferroptosis	B16	No	[38]
oncolytic vaccinia virus	Immunotherapy				
MiR-21-3p-AuNp	Nanoparticles	Ferroptosis	A375, A2058, WM793B	Yes	[48]
ICG001	Wnt inhibitor	Ferroptosis	A375, A2058, B16F10	Yes	[47]
RSL-3	GPX4 inhibition	Ferroptosis	A375, G-361	Yes	[13,41,42]
			A375, A2058, WM793B	Yes	[48]
			B16F10	Yes	[78]
			MeWo, A2058, SK-MEL-5, SKMEL-24, C8161, CHL-1, A375	No	[35]
HGF	Hydrophilic nanoparticle with polyphenol-iron and GW4869 inhibitor loaded	Ferroptosis	B16F10	Yes	[77]
Fe@OVA-IR820	Nanovaccines	Ferroptosis	B16-OVA	Yes	[36]
Curaxin CBL0137	Molecular compound	Necroptosis	B16F10, B16OVA, YUMMER1.7	Yes	[85]
MLKL-mRNA	Nanoparticles	Necroptosis	B16	Yes	[84]
AZD1775	WEE1 kinase inhibitor	Necroptosis	B16	Yes	[91]
GSK2334470	PDPK1i	Pyroptosis	WM1361A, WM1366	Yes	[25]
trametinib	MEKi				
Raptinal	Caspase-3 activator	Pyroptosis	D4M3.A, YUMM1.7, A375, WM35, WM793, and 1205Lu	Yes	[24]
AUY-922 PI-103 DHP1808	Hsp90i pan-PI3Ki	Pyroptosis	A375	No	[92]
BGB324	ALK inhibitor	Ferroptosis + autophagy + apoptosis	A375, WM1366, Melmet 1	Yes	[93]
Iridium(III) complex Ir-pbt-Bpa	PDT	Ferroptosis	A375, B16F10	Yes	[59]
PpIX-PSilQ NPs	Protoporphyrin IX-based PSilQ PLATFORM	Ferroptosis + apoptosis	A375	Yes	[94]
Etoposide	Chemotherapy drug	Pyroptosis	A375, D4M3.A, YUMM1.7	Yes	[22,53]
Doxorubicin	Chemotherapy drug	Pyroptosis	A375, D4M3.A, YUMM1.7	Yes	[22,53]
PLX4720	BRAFi	Pyroptosis	D4M3.A, YUMM1.7	Yes	[22]
PD0325901	MEKi				
RSL3 Lorlatinib	GPX4 inhibition ALK inhibitor	Ferroptosis	A375, A2058	Yes	[95]
Salinomycin	Plasma-activated infusion	Necroptosis	A2058	Yes	[96]
Compound 38	Pleuromutilin derivative	Necroptosis	A375, B16F10	Yes	[97]
Actinomycin-D	Chemotherapy drug	Pyroptosis	A375	No	[53]
NIR-PMCs	Nanoparticles	Ferroptosis	A375, B16, B16F10	Yes	[18]
Cu-BTC@DDTC	Nanoparticles	Ferroptosis	B16F10	Yes	[98]
zVAD-fmk	Caspase inhibitor	Necroptosis	B16F10	No	[57]
N-TiO2NPs	Nanoparticles	Necroptosis + autophagy	A375	No	[58]
Soyauxinium chloride	Natural compound	Apoptosis + ferroptosis + necroptosis	A2058, SK-MEL-505, SK-MEL-28, MEL-2A, B16F1, B16F10	No	[61]
Progenin III	Natural compound	Autophagy + ferroptosis + necroptosis	A2058, SK-MEL-505, SK-MEL-28, MEL-2A, B16F1, B16F10	No	[70]
Sanguilutine	Natural compound	Autophagy + necroptosis	A375	No	[69]
Salinomycin	Plasma-activated infusion	Necroptosis	A2058	Yes	[96]

## Data Availability

Not applicable.

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
