# Peer review of "Harnessing the Potential of Non-Apoptotic Cell Death Processes in the Treatment of Drug-Resistant Melanoma"

_ijms, 2023, doi:10.3390/ijms241210376_

Round 1

Reviewer 1 Report

This review deals with an overview of the relevance of non-apoptotic cell deaths in melanoma, focusing specifically on their molecular mechanisms and therapeutic implications.

General comments:

The topic of the review is of current importance, the review topic is covered with satisfactory completeness and references are appropriate.

Despite this, several concerns remain, especially about paragraph 4. In particular:

-                  the title of paragraph 4 covers only the topic of paragraph 4.4. I suggest changing the title of paragraph 4, keeping more general on the therapeutic implications.

-                  Referring to the first point, I suggest changing the title of the review not limiting it to the therapeutic implications in overcoming resistance, but keeping it more generally on the therapeutic implications. It may be “The emerging non-apoptotic cell deaths in melanoma: their molecular mechanisms and therapeutic implications”, or something similar.

-           the organization of paragraph 4 is confusing and, often, redundant. It needs to be reorganized taking into account the type of cell death (e.g. targeting ferroptosis, etc), reflecting paragraph 2, or the biological activity of drugs (e.g. anti-tumor activity as single agents, synergizing with other therapies, overcoming drug resistance, delaying the development of drug resistance).

-           Table 1 should be reorganized following the structure of paragraph 4.

Specific comments:

-           Figure 1 is not cited in the text

-           Line 36: delete decades

-           Lines 38-39. Please rewrite the sentence indicating that targeted therapy and immunotherapy are the most used for treating melanoma patients.

-           Line 39: what means monotherapy? The authors maybe refer to immunotherapy

-           Line 42: The abbreviation of programmed cell death is (PCD). Correct it through the text

-           Line 46: after “others” replace “,” with “.”  

-           Line 86: The sentence seems not referred to ref 17. Please verify it.

-           line 222-223: Please rearrange the sentence to avoid starting a paragraph with "for example"

-           Line 231: Correct “casepase-8”

-           Line 255: The abbreviation of extracellular matrix is ECM, please correct

-           Line 276: What means “…DCs in melanoma cells”? Please correct

-           Lines 518-522: These two sentences are out of the topic of the 4.5 paragraph. Please move them to future perspectives or conclusions.

An extensive editing of English language is required. Some sentences are difficult to understand  

Reviewer 2 Report

The review manuscript is well written and organizer. The topic is interesting and up-to-date. The Authors did not refer to other similar papers, however, this manuscript presents somehow another point od view.

Specific comments:

1. I recommend to change programmed cell death to regulated cell death. The Authors use RCD abbreviation. 

2. Table should be either extended or another one should be prepared to show clinical trials evaluating compounds that induce non- apoptotic cell deaths.

Round 2

Reviewer 1 Report

The revised manuscript is acceptable for publication